# The Impact of Probiotic *Bacillus subtilis* on Injurious Behavior in Laying Hens

**DOI:** 10.3390/ani12070870

**Published:** 2022-03-30

**Authors:** Sha Jiang, Jia-Ying Hu, Heng-Wei Cheng

**Affiliations:** 1Joint International Research Laboratory of Animal Health and Animal Food Safety, College of Veterinary Medicine, Southwest University, Chongqing 400715, China; jiangsha0527@swu.edu.cn; 2Department of Animal Sciences, College of Agriculture, Purdue University, West Lafayette, IN 47907, USA; hu165@purdue.edu; 3Livestock Behavior Research Unit, USDA-Agricultural Research Service, West Lafayette, IN 47907, USA

**Keywords:** chicken, social stress, injurious behavior, gut microbiota, serotonin, probiotic

## Abstract

**Simple Summary:**

Injurious behavior prevention is a critical issue in the poultry industry due to increasing social stress, leading to negative effects on bird production and survivability, consequently enhancing gut microbiota dysbiosis and neuroinflammation via the microbiota–gut–brain axis. Probiotics have been used as potential therapeutic psychobiotics to treat or improve neuropsychiatric disorders or symptoms by boosting cognitive and behavioral processes and reducing stress reactions in humans and various experimental animals. The current data will first report that probiotic *Bacillus subtilis* reduces stress-induced injurious behavior in laying hens via regulating microbiota–gut–brain function with the potential to be an alternative to beak trimming during poultry egg production.

**Abstract:**

Intestinal microbiota functions such as an endocrine organ to regulate host physiological homeostasis and behavioral exhibition in stress responses via regulating the gut–brain axis in humans and other mammals. In humans, stress-induced dysbiosis of the gut microbiota leads to intestinal permeability, subsequently affecting the clinical course of neuropsychiatric disorders, increasing the frequency of aggression and related violent behaviors. Probiotics, as direct-fed microorganism, have been used as dietary supplements or functional foods to target gut microbiota (microbiome) for the prevention or therapeutic treatment of mental diseases including social stress-induced psychiatric disorders such as depression, anxiety, impulsivity, and schizophrenia. Similar function of the probiotics may present in laying hens due to the intestinal microbiota having a similar function between avian and mammals. In laying hens, some management practices such as hens reared in conventional cages or at a high stocking density may cause stress, leading to injurious behaviors such as aggressive pecking, severe feather pecking, and cannibalism, which is a critical issue facing the poultry industry due to negative effects on hen health and welfare with devastating economic consequences. We discuss the current development of using probiotic *Bacillus subtilis* to prevent or reduce injurious behavior in laying hens.

## 1. Introduction

Numerous studies have revealed that the intestinal microbiota regulates host physiological homeostasis and behavioral exhibition in response to stress and related neurological disorders via the microbiota–gut–brain (MGB) and microbiota–gut–immune (MGI) axes in humans and other mammals [1,2,3,4,5,6,7,8,9,10]. Stress has a major impact on gut physiological homeostasis, leading to increased intestinal permeability (gut dysbiosis), subsequently causing neuroinflammation [11] and affecting the clinical course of neuropsychiatric disorders and behavioral exhibition including aggression [12,13,14,15,16]. Gastrointestinal diseases such as inflammatory bowel disease and irritable bowel syndrome have been linked to psychological disorders [17,18]. Fecal microbiota transplantation has potential to treat neuropsychiatric disorders [3,19,20]. Probiotics (or named direct-fed microbials) have been defined as: “live microorganisms which when administered in adequate amount confer a health benefit on the host” [21,22]. Numerous probiotics such as *Lactobacillus plantarum* and *Bifidobacterium lactis* have been used as dietary supplements or functional foods with therapeutic potential to target gut microbiota for the prevention or potential therapeutic treatment of mental diseases including social stress-induced psychiatric disorders such as depression, anxiety, and schizophrenia [15,19,23,24,25,26,27,28,29,30]. Similar function of the probiotics via modification of the gut microbiota may present in poultry.

Some management practices used in the poultry industry such as hatched under commercial conditions, reared in conventional cages or at high stocking density, mixed with strangers, and transported between housing facilities may cause stress in birds, leading to abnormal behavior such as injurious pecking (aggressive pecking and severe feather pecking) and cannibalism [31,32,33]. Injurious behavior prevention is a critical issue in the poultry industry due to the negative effects on bird production and survivability with devastating economic consequences [34,35,36,37,38]. Beak trimming (BT), the removal of 1/3 to 1/2 of a beak using hot blaze or infrared, is a routine husbandry procedure practiced in laying hens to prevent or reduce injurious behavior [39,40]. However, BT causes tissue damage, which may increase somatosensory sensitization of the damaged nerve tissues, resulting in pain (acute, chronic or both) in the trimmed birds [31,39,40]. Both enriched-caging and free-range production systems have been developed to meet the chickens’ ‘natural’ behavioral needs, however, injurious behavior still occurs within the flocks, and other management strategies must be implemented [32,41,42,43]. With the growing public concern for laying hen health and welfare, there is a pressing need to identify and develop alternatives to BT. This review focused on the recent findings to provide an overview of the feasibility of using probiotics as a management strategy to inhibit or reduce these injurious behaviors in laying hens.

## 2. Production Environments and Related Stress in Commercial Laying Hens 

Chickens as well as other farm animals are constantly selected by both nature (natural selection) and humans (artificial selection). During selection, the animals’ biological and behavioral characteristics have been constantly changed [44,45]. The process is affected by multiple factors including their surrounding environments, by which the animals have been selected for increased fitness (that is survival and reproductive success) over generations. 

Commercial chickens have been selected for production (laying hens for eggs and broilers for meat) to meet the increasing demand for poultry products [46,47]. The consumption of chicken meat and eggs represents cheap, healthy, and quality protein sources in human nutrition globally. However, the breeding programs may subject chickens to physiological dysfunction and immunosuppression by simply focusing on reproduction and or growth rates [48,49], subsequently increasing susceptibility to metabolic disorders and management-associated stressors [50,51]. For example, a laying hen produces approximately 310 eggs annually with a low feed consumption of just 110 g per day [52]. The extreme selection for one trait (production) could affect other biological traits, causing negative impacts on animal health and welfare such as aggression and related injurious behavior [53]. Selection for production increases aggression as, from an evolutionary point of view, aggression in animals is a natural behavior associated with competition to deal with life-threating situations affecting an individual’s survival, growth, and reproductive success within a group [54,55]. Controversially, selection based on hen behavior may reduce feather pecking, but it may result in an unfavorable correlated selection response, reducing egg production [56].

Currently, the conventional (battery) cage system is the most common housing facility for laying hens in the United Sates (U.S.), which was estimated to be 70.7% of the table egg layer flocks (approximately 231.7 million laying hens) at the end of 2020 [57]. Typically, commercial laying hens are housed in groups ranging from five to nine birds per cage or greater at a density of 67–86 in^2^/hen, starting at about 18 weeks of age. The high stocking density of hens and limited space for hens to display their “natural” behavior (such as foraging, exploration, perching, and nesting) negatively impact their welfare status, resulting in a chronic state of stress [58]. One of the possible strategies to improve hen health and welfare is to modify their rearing environments, and several alternatives to the conventional cage system have been developed such as enriched cage system (consisting of a nest, litter bath or scratch area, perches, and abrasive strip) and cage-free systems with or without outdoor access such as aviaries (single- and multiple-tiered) [59]. Although hens housed in the enriched cage system and non-cage systems seem to be possible ways to improve their welfare by displaying some degree of “natural” behavior such as nesting, roosting, and scratching [60,61], there is a high risk of increased exhibition of injurious behavior (feather pecking, aggression, and cannibalism) resulting from large group sizes and social instability [41,42,43,62,63]. Social stress and associated injurious behavior are major concerns in all current housing environments including cage and cage-free systems [41,42]. 

## 3. Injurious Behavior in Laying Hens

Chickens as well as other farm animals were domesticated from wild animals several thousand years ago. During domestication and subsequent artificial selection (breeding), the selected animals have continuously had their physiological stress response (adaptive capability, i.e., effect of reactive ability on growth, metabolism, and reproduction) and behavioral profile (social and emotional exhibitions as well as cognitive ability) changed to ‘fit’ the given environments and related management practices. However, not all individuals and species (phenotype and or genotype) of animals have equal capability to adapt to their environments or to modify their physiological and behavioral characteristics in response to environmental challenges [64]. Consequently, animals that exhibit less adaptation to their rearing environment may have a risk of poor health and welfare by reducing fitness [65]. 

### 3.1. Feather Pecking 

Feather pecking in laying hens is a behavior performed by birds pecking repetitively to conspecifics. It includes two categories: gentle and severe feather pecking [66], driven by different motivational systems [67]. Gentle feather pecking, repeated pecking at the tips and edges of feather without removal of the feather from the receivers, has been considered as a common behavior related to social discrimination and exploration [68] without association of severe feather pecking [69]; others have suggested that gentle feather pecking acts as a precursor of severe feather pecking and is associated with plumage damage [66]. Severe feather pecking, forcefully pecking at and pulling of feather, could be redirected foraging behavior [66,70], which is significantly increased if the hens’ foraging motivation cannot be fulfilled [71], if a suitable substrate is not provided [72,73] or removed [74]. A recent study indicates that feather pecking is a more complex reward and motivation procedure rather than redirected food-related foraging behavior, and is involved in multiple factors including motivation, foraging, exploration, and food selection [73]. Feather pecking, especially severe feather pecking, is a serious health and welfare problem in the poultry egg industry [37], which could affect up to 80% of birds in current housing systems [43,75]. In addition, feather pecking is associated with social stress-induced fearfulness [36]. Feather pecking in extreme cases leads to cannibalistic pecking, removing, and eating flesh from the victims, which can be further enforced via the gut–brain reward system (the central serotonergic and dopaminergic systems) [76,77], and spreads among the conspecifics (as a socially transmitted learning behavior) [78], leading to death. Cannibalistic pecking can be also induced by aggressive pecking following skin damage.

### 3.2. Aggressive Pecking and Cannibalism

Aggression within a group is to establish a dominance hierarchy when the animals are first brought together in a common environment [55,79]. In chickens, aggressive pecking, directly to the head and comb area, occurs to establish hierarchy within a group or in response to stress-induced social instability during rearing practices such as mixing individuals with unfamiliar birds during transfer from grower to layer facilities [80]. From an evolutionary perspective, aggression (also called combativeness) in animals is related to survival, growth, and reproduction [54,80,81]. Based on the natural selection theory, an animal’s productivity is correlated with its competitive ability. Dominance research has revealed a shift from a focus on species characteristics to the modern recognition of rich inter-individual variation (the behavioral and physiological phenotypes) [82]. Traditional techniques used for the selection of animals for breeding are primarily based on individuals with great targeted biomarkers (characteristics), mostly focused on productivity or profitability, resulting in a high risk of impaired ability to cope with their environment and biological problems, increasing competition and aggression [83]. In addition, based on the “frustration-aggression hypothesis”, aggression is a predictable reaction to external stimuli such as ambient conditions [84,85]. When restrictive environments such as conventional cages do not allow chickens to perform their natural behavior, they enter a state of frustration, with stress reactions and increased aggression [86,87]. In poultry, for example, egg production may have been increased through breeding selection while potentially injurious feather pecking and cannibalism have concurrently increased. For example, through more than 30 years of selection, egg production has been increased significantly in a former commercial Dekalb XL strain reared in conventional cages, whereas mortality due to aggression and cannibalism in non-beak trimmed hens had also increased about 10-fold [88,89] (Figure 1). 

### 3.3. Management and Beak Trimming in Laying Hens

In response to growing public pressures relating to poultry welfare, the management practices of laying hens have been modified, and various methods have been used to prevent or reduce social stress and stress-induced injurious behavior. For example, reducing light intensity, modifying nutritive value or taste of diets [64], providing straw, grain, or pelleted diets [42,91], rearing dual-purpose hybrids [38], housing hens in floor-pens [72], development of enriched cages [92,93], and aviary systems [41]. These management strategies have certain positive effects on laying hen welfare, but none of them provides a guarantee of preventing these injurious behaviors entirely.

Beak trimming (BT, also termed debeaking, beak tipping, beak mutilation, and partial beak amputation) is a routine procedure practiced in the U.S. egg industry and in most non-EU countries for preventing or reducing social stress and related injurious behavior [52]. The process involves the use of either infrared beak treatment (at the hatchery) or hot-blade beak trimming (prior to 10 days of age) to remove a portion of the upper, or upper and lower mandibles (i.e., the remaining length is 2–3 mm from the upper beak distal to the nostrils) [52]. A chicken’s beak is a complex, functional, and highly innervated organ that contains a great number of various sensory receptors including mechanoreceptors, thermoreceptors, and nociceptors [39,94]. Worldwide, BT continues to solicit a great deal of debate pertaining to the relative impact of BT on bird welfare. While the bestowed benefits of lowered aggression, feather pecking, and cannibalism may indeed favor improved well-being during the laying cycle [93,95], there are considerable anatomical, physiological, and biochemical changes that occur in cut peripheral nerves and damaged tissues, resulting in pain (acute, chronic, or both) with a negative impact on the welfare of billions of chickens annually [39]. There is growing pressure from animal welfare and consumer groups advocating the banning of this practice and to develop alternatives [96]. 

Injurious behavior can be improved through genetic selection such as group selection in which social interaction is included [34,97,98]. However, there is no sign that breeders will be able to guarantee ‘non-peck’ layers any time soon. One possibility is to develop an alternative to BT that minimizes social stress and pain, thereby preventing injurious behavior and related damage, by which it increases the health and welfare of laying hens. Recent studies have indicated that modification of gut microbiota composition, as a potential method, to prevent or reduce stress-induced gut dysbiosis and related inflammation [99,100,101,102], by which it further affects the hosts’ social behavior via the MGB and MGI axes [103]. 

## 4. Gut Microbiota, Stress, Injurious Behavior, and the Microbiota–Gut–Brain Axis

Gut microbiota is the collection of a large community with highly diverse microorganisms that reside within the gastrointestinal tract (GIT) of chickens as well as warm blooded animals. Its function resembles an endocrine organ engaged in multiple pathways (biological systems) including metabolic, immune, endocrine, and neural regulations by integrating the signals received from the internal and external stimulations via the bidirectional communications between the gut and brain [6,104,105,106,107,108,109]. Maintenance of gut microbial balance is essential for chickens to keep their physiological and behavioral homeostasis, which is critical for optimal growth, reproduction, health, and welfare under the current poultry industry globally. To meet the continuously growing demand for human consumption, the current breeding programs and management practices (focused on high production and economic efficiency) may result in multiple stressors affecting chicken health and welfare [31,106,107,108,109,110,111] (Figure 2). 

Under normal rearing conditions, chickens adapted to rearing-related factors have a balanced gut microbiota composition, maintaining its optimal function in feed digestion, nutrient resorption, synthesis of biochemicals, and neural and immune regulation [112,113]. However, under certain conditions, these management-associated stressors have negative effects on the gut microbial structure and functions by (1) disrupting the commensal bacterial populations and colonization (the stability of the gut microbiota), thus reducing beneficial bacteria and increasing pathogenic variant (low-grade inflammation); (2) increasing pathogen survival and invasive capability (bacterial translocation to increase neuroinflammation); (3) disrupting the absorption of nutrients and minerals including calcium, a key bone mineral; (4) disrupting microbial neuroendocrine functions (producing several signaling molecules and neurochemicals including serotonin (5-HT); (5) disrupting the gut epithelial barrier, thereby increasing intestinal permeability causing the gut to leak certain bacteria and harmful substances into the bloodstream (leaky gut), resulting in inflammation and or infection; and (6) damaging epithelial cells, thus producing free radicals and reducing antioxidant efficacy (oxidative stress) [31,114,115,116,117,118]. These changes resulted from the gut microbiota alterations (imbalanced microbiota composition with dysfunction) that influenced host behavioral display and health status via the nerve systems (i.e., the vagus nerve, enteric nerve, and autonomic nervous system), hormone signaling, immune system, and microbial metabolites (such as short chain fatty acids) to regulate the function of the gut–brain and gut–immune axes (Figure 3). Intestinal bacteria, for example, are involved in tryptophan metabolism [119,120,121]. Tryptophan, a precursor of 5-HT, directly affects brain 5-HT synthesis as tryptophan can pass the brain–blood barrier (BBB) [122,123], being a direct link between the gut microbiota and brain [122]. Tryptophan has long been used to attenuate aggressive behavior, control stress, and modulate immune function in humans and several species of farm animals including chickens [124,125]. In chickens, both genetic and phenotypic feather peckers of divergently selected high (HFP) and low (LFP) feather pecking lines have lower plasma tryptophan concentrations compared to their non-pecking counterparts [126]. Tryptophan-enriched diet (neurodietary supplements) fed chickens have elevated serotonergic activity (5-HIAA/5-HT ratio) in the hypothalamus, which results in a decreased stress response accompanied by a significant reduction in cortisol levels when exposed to social-mixing related stress [127,128,129]. The hypothalamic–pituitary–adrenal (HPA) axis is functionally involved in the pathophysiology of many neuropsychiatric disorders including major depressive disorder and cognitive dysfunction [130]; and tryptophan hydroxylase 2 (TPH2), rate-limiting enzyme of 5-HT synthesis in the brain, has been used as a therapeutic target for psychiatric disorders [131,132]. It has been proposed that the potential novel strategies of psychotherapy aim at increasing tryptophan concentrations through restoring the normal gut microbiota composition and intestinal homeostasis to prevent or reduce stress-induced abnormal behavior in humans [127,133] and various animals including laying hens [125,134]. 

Gut microbial alteration in humans is associated with depression, anxiety, and neuropsychiatric disorders caused by various neurodegeneration or neuroinflammation [11,135,136,137,138,139]. Similarly, the gut microbiota (microbiome) in chickens affect their health and emotion, memory, social, and feeding behavior [102,132]. Breeding-induced changes of gut microbiota (composition and or diversity) have been recognized as the major reason causing the changes in behavior, such as anxiety-like behavior in quails and social and feeding behavior in chickens and turkeys [140]. For example, selection for digestive efficiency in chickens causes the differences in gut microbiota compositions co-localizing with loci involved in feeding behavior, consequently, the most efficient birds have great feed intake with less fear [141]. In addition, feather pecking in laying hens has been proposed to be a consequence of the gut–brain axis dysregulation. Meyer et al. [142] reported that there were differences in intestinal microbial metabolites between HFP and LFP birds, which could impact on the function of the gut–brain axis. Recent studies have further revealed that the gut microbiota diversity is different in layer strains divergently selected HFP and LFP lines [143,144]. High feather pecking birds have a higher genera of *Clostridiales* but lower *Staphylococcus* and *Lactobacillus* compared to LFP birds [143,144]. Early-life homologous microbiota transplantation from adult HFP and LFP birds (i.e., receiving microbiota from the same line) influences the active behavior of recipient birds with long-lasting effects on developing feather pecking through the regulation of the serotonergic and immune systems [145]. In one of our recent studies, the chicks (a third line) that received cecal microbial transplantation (CMT) from line 6_3_ adult birds (a gentle line) displayed significantly less aggressive behavior (during paired behavioral test) with higher concentrations of brain serotonin than the chicks that received CMT from line 7_2_ adult birds (an aggressive line) and the controls (orally administrated saline) [146]. This suggests that aggression in chickens, similar to humans [84], could be prevented or reduced by the modification of gut microbiota composition and function. Similarly, fecal microbiota transplantation from aged broilers to young broilers daily from day 5 to day 12 altered recipient birds’ behavior, intestinal morphology, and gut microbiota composition [147]. The recipient birds exhibited lower fearfulness during both the novel arean test and vigilance test (a predator model) than the controls (orally administrated saline). The recipient birds also had a higher relative abundance of *Lentisphaeae* at the phylum level, while a lower *Megamonas* at the genus level, with increased activities of the glutamatergic synapse and N-glycan biosynthesis pathways revealed by the functional capabilities of microbial community analysis [147]. In addition, there were biological differences in gut motility (contraction velocity and amplitude) in chickens, which was corrected with the distinct feather pecking phenotypes [148]. In contrast, a recent study reported that the differences in gut microbiota compositions between the divergently selected HFP and LFP lines were not associated with feather pecking and antagonistic behavior due to there not being significant differences in microbial abilities [149]. Thus, the current results provide insights into understanding the function of gut microbiota in the regulation of stress response and injurious behavior in poultry, which are summarized in Table 1.

## 5. Probiotics, *Bacillus subtilis*-Based Probiotics, Social Challenge-Induced Aggression

Probiotics are commensal bacteria (“direct-fed microbials”, DFM) that offer potential health beneficial effects to the host’s stress response (acute, chronic or both). Several commercial probiotics have been used in poultry production [150,151], and numerous studies have shown that probiotics aid chickens in adapting to their environment and improving their health and welfare by: (1) altering the microbiota profile with beneficial bacteria to prevent the growth of pathogens and to compete with enteric pathogens for the limited availability of nutrient and attachment sites; (2) producing bacteriocins (such as bacteriostatic and bactericidal substances) with antimicrobial function and short chain fatty acids to regulate the activity of intestinal digestive enzymes and energy homeostasis; (3) modulating gut and systemic immunity; (4) restoring the intestinal barrier integrity preventing pathogens from crossing the mucosal epithelium; (5) stimulating the endocrine system and attenuating stress-induced disorders of the HPA and/or sympathetic-adrenal-medullary (SMA) axes via the gut–brain axis; (6) inducing epithelial heat shock proteins to protect cells from oxidative damage; and (7) synthesis and secretion of neurotransmitters such as 5-HT and tryptophan [104,107,113,115,152] (Figure 3). 

It has been stated in humans and non-human primates that the gut microbiota have potential effects on their hosts’ aggressive behaviors and anxiety symptoms [153,154,155]. In rodent studies, germ-free (GF) animals with exaggerated HPA responses to social stress can be normalized by certain probiotics [126,156,157]. In addition, probiotics have successfully attenuated anxiety and depressive behaviors in rat offspring separated from their mother [158,159] and the obsessive-compulsive-like behaviors in mice [160,161]. These results support the psychobiotics theory [15,162] (i.e., a special class of probiotics (beneficial bacteria) delivering mental and cognitive health benefits (such as anxiolytic and antidepressant effects) to individuals) and provide a potential to use probiotics as a biotherapeutic strategy for improving a host’s mental and cognitive function in humans and other animals including chickens [163,164,165,166,167,168,169,170]. Probiotics may have similar effects on chicken behavior due to the human–animal transmission occurs during the evolution and ecology of gastrointestinal microbial development (the host–microbial coevolution) [171,172]. Several probiotics have been used in preventing injurious damage in poultry. For example, probiotic *Lactobacillus rhamnosus* JB-1 supplementation (5 × 10^9^/mL in drinking water provided from week 19 to week 28) reduces chronic stress (social disruption, physical and manual restraint, and blocking nest boxes and perch usage applied from week 24 to week 26) induced feather pecking and cecal microbiota dysbiosis, along with increased T cell populations in the spleen and cecal tonsils of adult chickens regardless of the genetic lines (HFP and LFP lines) [173] (Table 2). Probiotic *Lactobacillus rhamnosus* supplement (applied from day 1 to week 9) also counteracted stress-induced decrease in T cells, along with a short-term (from week 10 to week 13) increase in plasma tryptophan and the TRP:(PHE + TYR) ratio (from week 14 to week 15), but without effects on feather pecking in pullets [126]. The TRP:(PHE  +  TYR) ratio has been used as an indicator of the competition between tryptophan and other amino acids for uptake across the BBB [174]. In addition, the number of feather pecking bouts was positively correlated with intestinal contraction velocity and amplitude in peckers, which can be modulated by administrated *L. rhamnosus* [148]. *Lactobacillus-based probiotic* supplements also reduced stress-associated immobility behavior in rodents during the forced swine test [175]. Parois et al. [176] also reported that probiotic *Pediococcus acidilactici* reduced fearfulness in selected short tonic immobility birds, indicated by a short immobility during the tonic immobility test via regulation of the MGB axis. Reduced fearfulness was also found in a synbiotic study [177]. It consisted of a probiotic (*Enterococcus faecium*, *Pediococcus acidilactici*, *Bifidobacterium animalis*, and *Lactobacillus reuteri*) and a prebiotic (fructooligosaccharides). The synbiotic fed broilers had a shorter latency to make the first vocalization with a higher vocalization rate during an isolation test, and a greater number of synbiotic fed birds reached the observer during a touch test. There results revealed a potential strategy to use probiotics to reduce stress response and stress-induced injurious behavior during poultry production. However, large gaps about probiotic functions in improving neuropsychiatric disorders remain, which are affected by multiple factors including the type of probiotic bacteria and duration and dosage of the intervention. 

### Bacillus subtilis

*Bacillus subtilis* is one of the three most common species of probiotic products in the U.S. [181] and has been used widely as a functional feed supplement such as in several dairy and non-dairy fermented foods for improving human health and well-being [182,183,184,185]. Similarly, *Bacillus subtilis*-based probiotics have been used as antibiotic growth promoter alternatives in poultry [186,187,188,189]. *Bacillus subtilis* are spore-forming bacteria. They are heat stable, low pH-resistant (the gastric barrier), and tolerate multiple environmental stressors [190,191]. Several mechanisms of actions of *Bacillus* spp. have been proposed: regulating intestinal microstructure [192] and digestive enzymes [193,194]; synthesizing and releasing antimicrobial and antibiotic compounds [187]; increasing immunity [193,195,196], and neurochemical activities including 5-HT [197,198,199] as well as affecting animal behavior [198] following various stressors. For example, in response stimulations, *Bacillus subtilis* alleviates oxidative stress, provokes a specific biological response, and improves the mood status of hosts via the gut–brain axis [79,200]. In addition, *Bacillus subtilis* can overproduce L-tryptophan [201,202,203], and consequently increase 5-HT in the hypothalamus [204]. Tryptophan functions as an antidepressant and anti-anxiety agent [204,205,206,207] and eliminates nervous tension in mice [208,209]. In one study, chickens were used as an animal model to assess whether dietary supplementation of the probiotic *Bacillus subtilis* reduced aggressive behaviors following social challenge [178].

Chickens, as social animals, show fear, depression, and or anxiety in novel environments [172] and show aggression toward others for establishing aa social dominance rank in unfamiliar social groups [210,211,212], which is similar to the rodents used in human psychopharmacological studies [213,214,215]. The paired social ranking-associated behavioral test used in this study [206] has been routinely performed in chicken behavioral analysis [201,216,217]. The rationale of the test is similar to the resident-intruder test, which is a standardized method used in rodents for detecting social stress-induced aggression and violence [180,218,219,220]. 

In the study [208], the role of the probiotic *Bacillus subtilis* on the aggression in hens of the Dekalb XL (DXL) line was examined. One-day-old female chicks were kept in single-bird cages [178]. The hens at 24-weeks-old were paired based on their body weight for the first behavioral test (pre-probiotic treatment, day 0) in a novel floor pen. To determine the dominant individual per pair, behaviors were video-taped for 2 h immediately after the release of two hens simultaneously into the floor pen. After the test, the subordinate and dominant hens were fed the regular diet or the diet mixed with 250 ppm probiotic (1.0 × 10^6^ cfu/g of feed) for two weeks, respectively. The probiotic contained three proprietary strains of *Bacillus subtilis* (Sporulin^®^, Novus International Inc., Saint Charles, MO, USA). After the treatment (day 14), the second aggression test was conducted within the same pair of hens. The injurious behaviors were detected and analyzed (Table 3). 

The results indicated that compared to their initial levels at day 0, the levels of threat kick were reduced (Figure 4A. *p* = 0.04), the frequency of aggressive pecking tended to be lower (Figure 4B. *p* = 0.053), and the levels of feather pecking was reduced but without statistical significance (Figure 4C. 60%. *p* = 0.33) in probiotic fed dominant hens. There was no change in injurious behaviors in the regular diet fed subordinate hens between day 0 and day 14 (Figure 4A–D). The behavioral changes in probiotic fed dominant hens were correlated with the changes in blood 5-HT concentrations. Post-treatment (day 14), plasma 5-HT levels were reduced toward the levels of the controls (subordinates) in the probiotic fed dominant hens (Figure 5. *p* = 0.02) compared to their related levels prior to treatment (day 0). Similarly, the effects of probiotic dietary supplements on behavior have been found in turkeys [180]. The turkey poults fed probiotic *Bacillus amyloliquefaciens* had increased feeding frequency and duration with decreased distress call and aggressive behavior.

The similar relations between reduced aggressive behavioral exhibition and blood 5-HT concentrations were identified in our previous studies [53,89], genetic selection for prevention of social stress-induced feather pecking, and aggression. Compared to MBB mean bad birds (MBB), kind gentle birds (KGB) had lower blood 5-HT concentrations as well as lower concentrations of blood dopamine (DA) and corticosterone (CORT) and a lower heterophil/lymphocyte (H/L) ratio, a stress marker, with lower frequency of injurious pecking [53,91] (Table 4). Bolhuis et al. [221] also reported that peripheral serotonin activity reflected the predisposition to develop severe feather pecking in laying hens. Similarly, individuals with a lower blood 5-HT level that showed less aggressiveness were found in humans [222,223,224] and canine [225] while an elevated level of blood 5-HT has been revealed in patients with aggressive behavior [226,227] and in aggressive teleost fish [228]. These results provide evidence for serotonergic mediation for aggressive behavior and stress coping strategy; and chicken aggression can be reduced or inhibited by probiotic supplementation by directly or indirectly regulating the serotonergic system.

Whether the changes in blood 5-HT levels in probiotic fed dominant hens represent a similar change in 5-HT concentrations in the brain is unclear as 5-HT cannot pass the blood–brain barrier and is regulated differently between brain neurons and peripheral tissues [229]. The plasma 5-HT is synthesized mainly by the enterochromaffin (EC) cells (also known as Kulchitsky cells), types of enteroendocrine and neuroendocrine cells, of the gut and stored in the platelets [230]. However, it has been proposed that platelet 5-HT uptake is a limited peripheral marker of brain serotonergic synaptosomes [229]. *Lactobacillus plantarum* strain PS128, a dietary probiotic that causes an increase in the levels of striatal 5-HT as well as DA, is correlated with improving anxiety-like behavior in germ-free (GF) mice [230]. Similar results have been obtained from our current studies [178,179]. In another study, chickens (broilers) were fed *Bacillus subtilis* from day 1 to day 43. The results indicate that *Bacillus subtilis* fed chickens had higher levels of 5-HT in the raphe nuclei and lower levels of norepinephrine (NE) and DA in the hypothalamus compared to the controls fed a regular diet [230]. Probiotic fed chickens also had improved skeletal traits (bone mineral density, bone mineral content and robusticity index). In one heat stress (32 °C for 10 h) study, *Bacillus subtilis* fed chickens (broilers) had lower heat stress-related behaviors including panting and wing spreading and inflammatory response in the hypothalamus compared to the controls [203]. Further studies, however, are needed to examine how the correlations present between injurious behavior and peripheral and or brain 5-HT in probiotic fed chickens.

## 6. Conclusions and Perspectives

Injurious behavior is a critical issue facing the poultry industry due to increasing social stress, leading to negative effects on bird production and survivability. Numerous studies have revealed that enteric microbiota play a critical role in the hosts’ response to acute and chronic stress. Social stress-induced changes of the gut microbiota lead to inflammation and ‘leaking out’ of bacterial metabolites, affecting brain function, especially the function (activation) of both the HPA axis and the SMA axis. These changes, consequently, negatively affect the physiological and behavior homeostasis, leading to mental disorders with abnormal behaviors including aggression. Several recent studies suggest that dietary inclusion of probiotics such as *Bacillus subtilis* have positive effects on reducing agonistic behavior in laying hens through the modification of the serotonergic system. The novel approach could be transferred directly or indirectly to other species of farm animals that are subjected to painful husbandry procedures such as the dehorning of calves and teeth-clipping of piglets to prevent body injuries.

## Figures and Tables

**Figure 1 animals-12-00870-f001:**
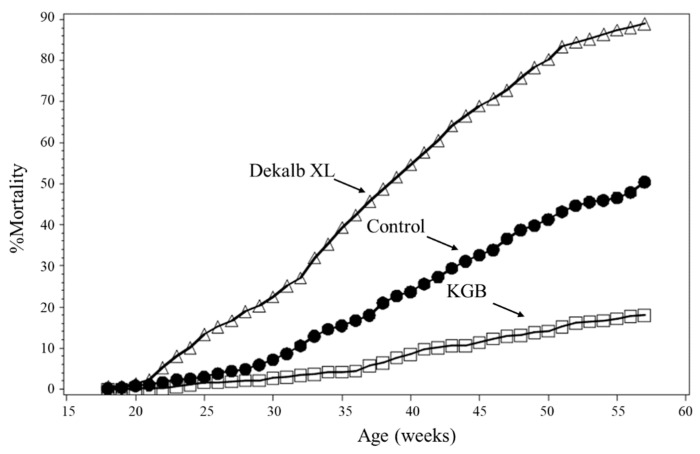
Percent mortality of the birds. Compared to the birds from both Dekalb XL (a commercial line) and the control (non-selected) lines, the birds from the selected KGB (kinder gentle bird) line had the lowest mortality [90].

**Figure 2 animals-12-00870-f002:**
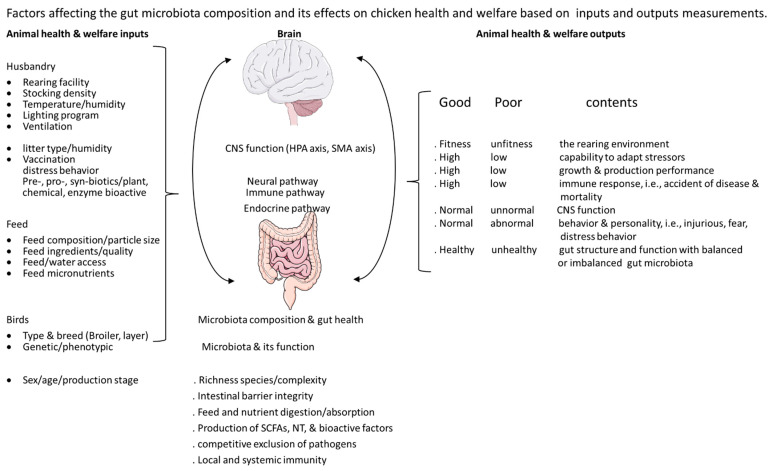
Factors affecting the gut microbiota composition and the mechanisms of its effects on chicken health and welfare via the bilateral gut–brain connections based on inputs and outputs animal welfare measurements (modified according to Chen et al. [106] and Shehata et al. [107]). Figure was created with BioRender.com, accessed on 15 March 2022.

**Figure 3 animals-12-00870-f003:**
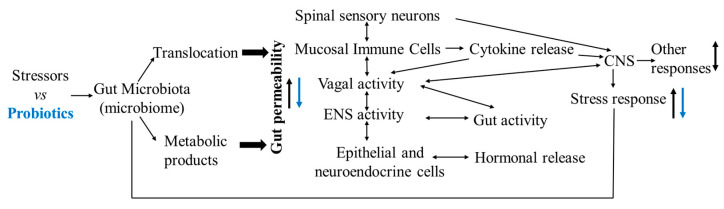
The effects of stressors and probiotics on the gut microbiota–host interaction. Stressors increase gut permeability, consequently increasing host stress response and stress-induced physiological and behavioral disorders via the neural, endocrine, and immune pathways. Probiotics reverse the stress-induced gut microbial disorders and recover the physiological and behavioral changes in hosts via multiple pathways (modified from Yarandi et al. [12]).

**Figure 4 animals-12-00870-f004:**
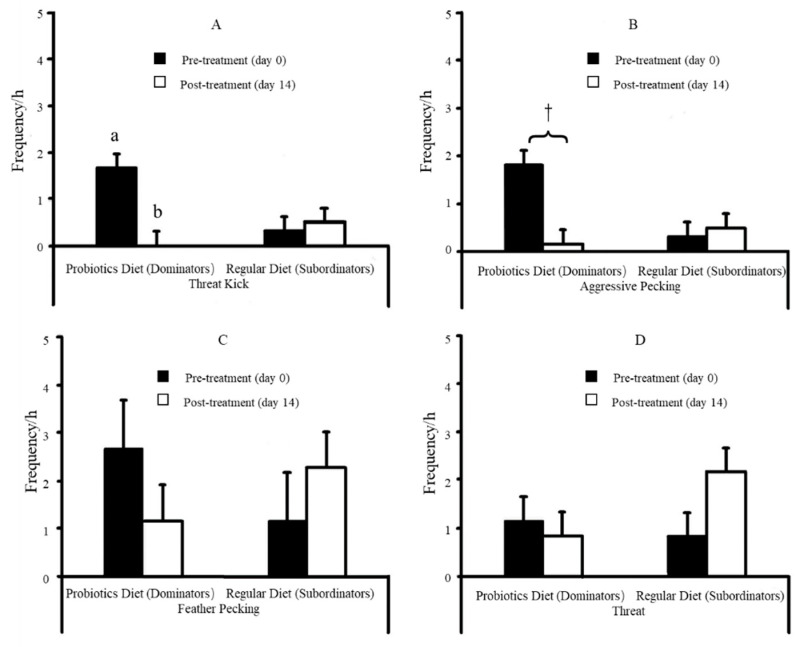
Frequency of aggressive behaviors at day 0 (pre-treatment) and day 14 (post-treatment) in probiotic fed hens and regular diet fed hens followed by the paired social test (*n* = 12). (**A**) The frequency of threat kick. (**B**) The frequency of aggressive pecking. (**C**) The frequency of feather pecking. (**D**) The frequency of threat. The exhibition of aggressive behaviors in the regular diet fed subordinates was not affected by treatment (*p* > 0.05, respectively), while the frequency of threat kick (*p* = 0.04) was reduced, aggressive pecking (*p* = 0.053) tended to be lower, and feather pecking was lower (60%, *p* = 0.33) in probiotic fed dominant hens post-treatment. Notes: The treatment effects on the measured behaviors were reversed between dominants and subordinates during the second social rank test. a,b Between the frequency at day 0 and day 14, least square means lacking common superscripts differ (*p* < 0.05); and † trend difference (0.05 ≤ *p* < 0.10). A modified copy from Hu et al. [178].

**Figure 5 animals-12-00870-f005:**
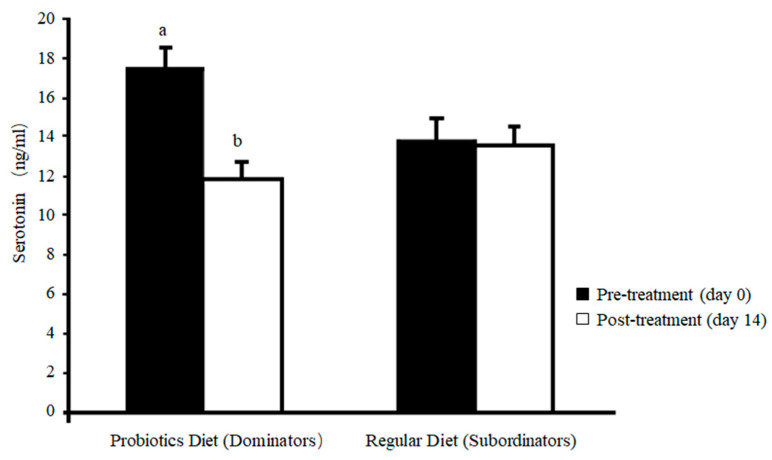
Plasma serotonin (5-HT) levels at day 0 (pre-treatment) and day 14 (post-treatment) in probiotic fed dominant hens and regular diet fed subordinate hens. The plasma 5-HT concentration was measured by using the high performance liquid chromatography (HPLC). Compared to subordinate hens, plasma 5-HT concentrations were higher in dominant hens at day 0 but without statistical difference (*p* = 0.24); the difference disappeared at day 14. Compared to the levels at day 0, blood concentrations of 5-HT were reduced in probiotic fed dominant hens at day 14 (*p* = 0.02) but were not in regular diet fed subordinate hens (*p* > 0.05). a,b Between the concentrations at day 0 and day 14, least square means lacking common superscripts differ (*p* < 0.05). A copy from Hu et al. [178].

**Table 1 animals-12-00870-t001:** A list of recent studies showing the relationship between intestinal microbiota composition and injurious behavior in laying hens.

Treatment	Effects	Conclusion	Reference
H & L birds of the HFP & LFP lines	Selection caused line’s differences in intestinal microbial metabolites.	These changes could relate to lines’ differences in behaviors via the MGB axis.	[142]
Birds of the HFP & LFP Lines	HFP birds had ↓cecal microbial beta diversity with ↑relative abundance of *Clostridiae* but ↓*Lactobaccillacae*.	Selection causes differences in the cecal microbial profile; cecal microbiota may involve in FP behavior.	[143]
Genotypes: Selected HFP & LFP lines	HFP birds had ↑genera of *Clostridiales*, ↓*Staphylococcus* & *Lactobacillus* in LM. HFP birds had ↑diversity & evenness for both cecal MAM & LM. HFP neutral birds had ↑genera of *Clostridiales*, ↓*Lactobacillus* with ↑diversity & evenness in LM but not MAM.	Feather pecking genotype but not phenotype affects LM composition; but the correlation between FP and microbiota composition remains to be elucidated.	[144]
Phenotypes: Feather peckers & neutrals			
HFP & LFP recipients received microbiota from the same line (HMT) during the 1st two weeks	HMT influenced immune characteristics in both lines. HMT influenced active behavior and peripheral serotonin in the LFP line. HMT without effects on gut microbiota composition, stress response, and FP.	Early MT may influence the development of FP due to its effects on FP-associated behavioral and physiological characteristics.	[145]
Birds of the HFP & LFP lines	HFP birds had ↓*genera Lactobacillus*, ↑*Escherichia* in ileum digesta, and ↓*Faecalibacturium* & *Blautia* in cecal digesta & mucoa.	Gut microbial composition and its functions are not associated with FP & antagonistic behavior.	[149]
DXL recipients received CMT from lines 6_3_, 7_2_ or saline orally from day 1 to day 10, then boosted once from week 3 to week 5	HFP birds had ↑tryptophan metabolism & lysine degradation in digesta & mucosa. There were no line effects on microbial abilities.		[146]
	6_3_-CMT recipients displayed less aggressive behavior during paired aggression test with higher concentrations of serotonin.	Early-life CMT has the potential to reduce aggressive behavior through the GMB axis.	

Line 6_3_: gentle birds; Line 7_2_: aggressive birds; CMT: cecal microbiota transplantation; FP: feather pecking, H: high feather-pecking birds; HFP: high feather pecking line; HMT: microbiota transplantation within the same line; L: low feather-pecking birds; LFP: low feather line; LM: luminal microbiota, MAM: mucosa-associated microbiota; MGB axis: the microbiota–gut–brain axis.

**Table 2 animals-12-00870-t002:** Non-comprehensive list of recent studies showing the effects of probiotics modifying gut microbiota on behaviors in poultry.

Birds/Treatment		Effects	Conclusion	Reference
Layers	*L. rhamnosus* (5 × 10^9^ CFU/mL) in-drinking water of stressed HFP birds, LFP birds, & a unselected pullets	Pecks (phenotypic and genotypic) had lower plasma TRP. *L. rhamnosus* caused a short-term increase in plasma TRYP and the TRP:(PHE + TYR) ratio and all subsets of T cell proportions.	A transient effect on the immune and TRP catabolism with minimal changes in behavior in pullets.	[126]
	*L. rhamnosus* (5 × 10^9^/mL) in-drinking water of chronic stressed adult HPF & LPF hens	*L. rhamnosus* prevented stress-induced FP. *L. rhamnosus* increased T cells in the spleen and cecal tonsil. *L. rhamosus* reduced cecal microbiota dysbiosis.	Reduces stress-induced FP; and improves hen welfare.	[173]
	*L. rhamnosus* (5 × 10^8^ CFU/mL) orally fed peckers & non-peckers	*L. rhamosus* caused ↑cecal contractions and their amplitude; It positively correlated with the number of FP of peckers.	Impacted gut motility with FP phenotypic effects.	[148]
	*B. subtilis* (1 × 10^6^ CFU/g) fed dominant & subordinate hens	*B. subtilis* caused ↓threat kick and ↓aggressive pecking during paired aggression test, and ↓plasma serotonin. *B. subtilis* caused ↓HS-associated behavior but ↑eating, foraging, standing, and walking. *B. subtilis* led to ↓hepatic IL-6, HSP70, cecal IgA & IgY but ↑hepatic IL-10.	Dietary probiotic could be a suitable strategy for controlling aggression in chickens.	[178]
Broilers	*B. subtilis* (1 × 10^6^ CFU/g) fed HS broilers		Reduces HS-induced inflammatory reactions via the microbiota-immune axis, while increases broilers to copy HS more effectively.	[179]
	* A synbiotic fed HS broilers	Snybiotic fed birds had a shorter latency to make the first vocalization, with higher vocalization rates during the isolation test and a greater number of birds reached the observer during the touch test.	The synbiotic can reduce the fear response and stress state of HS broilers.	[177]
Turkeys	*B. amyloliquefaciens* fed turkey poults	Probiotic increased the feeding time and decreased distress call and aggressive behaviors.	Probiotics regulates behavior in turkey poults via modulation of gut microbiota.	[180]
Quails	*P. acidilactici* (2.54 × 10^6^/g) fed STI & LTI quails	Probiotic reduces immobility duration of STI birds during TI test.	The probiotic affected host behavior and memory via the effects on gut microbiota	[176]

*B. amyloliquefaciens*: *Bucillus amyloliquefaciens*; *B. subtilis*: *Bacillus subtilis*; CMP: cecal microbiota transplantation; FP: feather pecking; HMT: homologous microbiota transplantation from the same line; HPF: high feather pecking birds; HS: heat stress; HSP: heat shock protein; IL: interleukin; *L. rhamosus*: *Lactobacillus rhamnosus*; IL: interleukin; LFP: low feather pecking birds; LTI: long tonic immobility quails; *P. acidilactici*: *Pediococcus acidilactici*; PHE: phenylalanine; STI: short tonic immobility quails; T: T lymphocytes; TI: Tonic immobility test; TRP: tryptophan; TYR: tyrosine. * Synbiotic consisted of a probiotic (*Enterococcus faecium*, *Pediococcus acidilactici*, *Bifidobacterium animalis*, and *Lactobacillus reuteri*) and a prebiotic (fructooligosaccharides).

**Table 3 animals-12-00870-t003:** Behavioral ethogram ^1^.

Behavior	Description
Feather Pecking	One bird pecking at feathers of another bird can be (a) gentle peck (nibbling or gentle pecking in which feathers are not removed or pulled) or (b) severe peck (vigorous pecking to feathers in which feathers are often pulled, broken, or removed).
Threat	One bird standing with its neck erect and hackle feathers raised in front of another bird.
Aggressive pecking	Forceful downward pecks directed at the head or neck of other birds
Threat Kick	One bird forcefully extending one or both legs such that the foot strikes another bird.

^1^ Modified from Hu et al. [178].

**Table 4 animals-12-00870-t004:** Selection-induced alterations in blood concentrations of dopamine, serotonin, and corticosterone in laying hens.

Lines	Corticosterone (ng/mL)	Dopamine (ng/mL)	Epinephrin (ng/mL)	Serotonin (ng/mL)	H/L Ratio ^2^ (×100)
KGB ^1^	1.87 + 0.19	0.59 + 0.08 ^a^	0.30 + 0.06 ^a^	11.8 + 0.07 ^a^	13.0 ^a^
MBB	1.49 + 0.21	2.42 + 0.76 ^b^	0.59 + 0.13 ^b^	14.3 + 0.06 ^b^	29.4 ^b^

^a,b^ Means within a column with different superscript are statistically different (*n* = 12, *p* < 0.05). ^1^ The KGB (kind gentle bird) and MBB (mean bad bird) lines were selected for high and low productivity and survivability resulting from cannibalism and flightiness, respectively. ^2^ Heterophil/lymphocyte ratio [90].

## Data Availability

Not applicable.

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
