# Peer review of "The Impact of Probiotic Bacillus subtilis on Injurious Behavior in Laying Hens"

_animals, 2022, doi:10.3390/ani12070870_

Round 1

Reviewer 1 Report

In general, the review is well written, and my comments aimed to increase the scientific soundness and clarity of it.

My concerns:

  1. Because the present review deals with the birds any references to human and mammals in general are superfluous. Please rewrite lines 35-51
  2. Therefore, chapter 4 is barely related to avian gut microbiota. It mostly describes relationships in the mammalian GIT.
  3. The same comments as above are partially true in relation to the chapter 5. Please focus on the birds instead of mammals.
  4. Line 166 – Please make sure that you have permissions form the publishers for reuse of figures 1-4 as well as tables 1-2.
  5. Line 220 – the phrase “anatomical disorders” is awkward. Anatomy is the branch of science concerned with the study of the normal structure of organisms and their parts. So any disturbances in anatomy must be called pathology.
  6. Line 403 – it is not clear what the authors found if they only reviewed the literature? The use of word “finding” without any experimental design is not justified.
  7. Line 31, 258, 314, 386, 390, 404 and references - Bacillus subtilis should be written in italic.

Author Response

  1. Because the present review deals with the birds any references to human and mammals in general are superfluous. Please rewrite lines 35-51

We have had the similar thoughts when we started to search relative articles for preparing the manuscript with the aim: if probiotics can reduce social stress and stress-related injurious behaviors in chickens. However, through the process, the most articles related to the aim are conducted in humans and experimental animals, especially in the areas using probiotics as potential therapeutic psychobiotics to treat or improve neuropsychiatric disorders and neurodegeneration-associated mental diseases. Although the gut microbiota effects on feather pecking in laying hens have been investigated (Birkl et al., 2018; van der Eijk et al., 2019, 2020), few studies have been conducted in investigating the potential use of probiotics to prevent or reduce social stress and stress-induced injurious behaviors in chickens. Based on the understanding, our study was the 1st investigation conducted in chickens (Hu et al. 2018; Wang et al., 2018; Mohammed et al., 2021). Recently, a few studies using probiotics to modify gut microbiota via the gut-brain axis to prevent feather pecking have been published, for example, use of Lactobacillus rhamnosus (van Staaveren et al., 2020; Mindus et al., 2021a,b) to reduce chronic stress-induced feather pecking via improving immune response and tryptophan catabolism in laying hens. The information has been included in the manuscript. Hopefully, the article with the aim focused on the effects of probiotics, gut microbiota, and their interactions on mental health will provide a new insight for developing an alternative to beak trimming in poultry.

References

Birkl P, Bharwani A, Kjaer JB, Kunze W, McBride P, Forsythe P, Harlander-Matauschek A. 2018. Differences in cecal microbiome of selected high and low feather-pecking laying hens. Poult. Sci. 97(9):3009-3014.

Hu J, Chen H, Cheng H.W. 2018. Effect of direct-fed microbials, Bacillus subtilis, on production performance, serotonin concentrations and behavioral parameters in a selected dominant strain of white leghorn hens. Int. J. Poult. Sci. 2018, 17, 106-115.

Mindus C, van Staaveren N, Bharwani A, Fuchs D, Gostner JM, Kjaer JB, Kunze W, Mian MF, Shoveller AK, Forsythe P, Harlander-Matauschek A. 2021a. Ingestion of Lactobacillus rhamnosus modulates chronic stress-induced feather pecking in chickens. Sci Rep. 11(1):17119.

Mindus C, van Staaveren N, Fuchs D, Gostner JM, Kjaer JB, Kunze W, Mian MF, Shoveller AK, Forsythe P, Harlander-Matauschek A. 2021b. L. rhamnosus improves the immune response and tryptophan catabolism in laying hen pullets. Sci Rep. 11(1):19538.

Mohammed A, Mahmoud M, Murugesan R, Cheng HW. 2021. Effect of a Synbiotic Supplement on Fear Response and Memory Assessment of Broiler Chickens Subjected to Heat Stress. Animals 11(2):427. van der Eijk JAJ, de Vries H, Kjaer JB, Naguib M, Kemp B, Smidt H, Rodenburg TB, Lammers A. 2019. Differences in gut microbiota composition of laying hen lines divergently selected on feather pecking. Poult. Sci. 98(12):7009-7021.

van der Eijk JAJ, Rodenburg TB, de Vries H, Kjaer JB, Smidt H, Naguib M, Kemp B, Lammers A. 2020. Early-life microbiota transplantation affects behavioural responses, serotonin and immune characteristics in chicken lines divergently selected on feather pecking. Sci Rep. 10(1):2750.

van Staaveren N, Krumma J, Forsythe P, Kjaer JB, Kwon IY, Mao YK, West C, Kunze W, Harlander-Matauschek A. 2020. Cecal motility and the impact of Lactobacillus in feather pecking laying hens. Sci Rep. 10(1):12978.

Wang WC, Yan FF, Hu JY, Amen OA, Cheng HW. 2018. Supplementation of Bacillus subtilisbased probiotic reduces heat stress-related behaviors and inflammatory response in broiler chickens. J. Anim. Sci. 2018, 96, 1654-1666.

  1. Therefore, chapter 4 is barely related to avian gut microbiota. It mostly describes relationships in the mammalian GIT.

The gastrointestinal system in chickens, similar to it in humans, harbors a large microbial community which plays an important role in their growth and health. Domestic chicken is a common model for human biological research including investigation of the function of gut microbiota in pathophysiological homeostasis and its underlying mechanisms due to that the human – animal transmission occurs during the evolution and ecology of gastrointestinal microbial development (Oakley et ail., 2012; Hamid et al., 2019; Hu et al., 2017). However, the most current available data regarding microbiota functionality are collected from the studies conducted in humans due to the significance of gut mcrobiome on human health. Although there are differences in gut microbiota composition between humans and chickens (Kollarcikova et al., 2020), the principles mechanisms of gut microbiota among warmblooded animals (humans, non-mammal animals, and other vertebral animals) are similar (Ellis et al., 2013; Nguyen et al., 2015; Kollarcikova et al., 2020; Ericsson and Franklin, 2021; Reese et al., 2021). Currently, the most gut microbiota studies in poultry are focused on production, health, and welfare with and without stress and immune challenges (Lee et al., 2022). The understanding the mechanisms of gut microbiota in humans and other mammals should provide critical information for the investigation of gut microbiota functions in chickens and other farm animals.

References

Ellis RJ, Bruce KD, Jenkins C, Stothard JR, Ajarova L, Mugisha L, Viney ME. 2013. Comparison of the distal gut microbiota from people and animals in Africa. PLoS One. 8(1):e54783.

Ericsson AC, Franklin CL. 2021. The gut microbiome of laboratory mice: considerations and best practices for translational research. Mamm. Genome. 32(4):239-250.

Hamid H, Zhang JY, Li WX, Liu C, Li ML, Zhao LH, Ji C, Ma QG. 2019. Interactions between the cecal microbiota and non-alcoholic steatohepatitis using laying hens as the model. Poult. Sci. 98(6):2509-2521.

Hu SL, Wang L, Jiang ZY. 2017. Dietary Additive Probiotics Modulation of the Intestinal Microbiota. Protein Pept. Lett. 24(5):382-387.

Kollarcikova M, Faldynova M, Matiasovicova J, Jahodarova E, Kubasova T, Seidlerova Z, Babak V, Videnska P, Cizek A, Rychlik I. 2020. Different Bacteroides Species Colonise Human and Chicken Intestinal Tract. Microorganisms. 8(10):1483.

Lee MD, Ipharraguerre IR, Arsenault RJ, Lyte M, Lyte JM, Humphrey B, Angel R, Korver DR. 2022. Informal nutrition symposium: leveraging the microbiome (and the metabolome) for poultry production. Poult. Sci. 101(2):101588.

Lei F, Yin Y, Wang Y, Deng D, Yu H, Li L, Xiang C, Wang S, Zhu B, Wang X. 2012. Higher-level production of volatile fatty acids in vitro by chicken gut microbiotas than by human gut microbiotas as determined by functional analyses. Appl. Environ. Microbiol. 78(16):5763- 5772.

Mosites M, Sammons M, Otiang E, Eng A, Noecker C, Manor O, Hilton S, Thumbi SM, Onyango C, Garland-Lewis G, Call DR, Njenga MK, Wasserheit, JM, Zambriski JA, Walson JL, Palmer GH, Montgomery J, Borenstein E, Omore R, Rabinowitz PM. 2017. Microbiome sharing between children, livestock and household surfaces in western Kenya. PLoS One. 12:e0171017.

Nguyen TL, Vieira-Silva S, Liston A, Raes J. 2015. How informative is the mouse for human gut microbiota research? Dis. Model. Mech. 8(1):1-16.

Oakley BB, Lillehoj HS, Kogut MH, Kim WK, Maurer JJ, Pedroso A, Lee MD, Collett SR, Johnson TJ, Cox NA. 2014. The chicken gastrointestinal microbiome. FEMS Microbiol. Lett. 360(2):100-12.

Reese AT, Chadaideh KS, Diggins CE, Schell LD, Beckel M, Callahan P, Ryan R, Emery Thompson M, Carmody RN. 2021. Effects of domestication on the gut microbiota parallel those of human industrialization. Elife. 10:e60197.

  1. The same comments as above are partially true in relation to the chapter 5. Please focus on the birds instead of mammals.

The similar reasons above are also related to the chapter 5.

  1. Line 166 – Please make sure that you have permissions form the publishers for reuse of figures 1-4 as well as tables 1-2.

Thanks for the reminder. Yes. We have the permission for use of the figures.

  1. Line 220 – the phrase “anatomical disorders” is awkward. Anatomy is the branch of science concerned with the study of the normal structure of organisms and their parts. So any disturbances in anatomy must be called pathology.

Agree with the reviewer. “Anatomical” has been replaced with “pathomorphological”.

  1. Line 403 – it is not clear what the authors found if they only reviewed the literature? The use of word “finding” without any experimental design is not justified.

In agree with the reviewer, “finding” has been removed.

  1. Line 31, 258, 314, 386, 390, 404 and references - Bacillus subtilis should be written in italic. Done as indicated.

Reviewer 2 Report

Dear Authors,

Thank you for submitting this paper to review. This is an interesting review of the impact of gut microbiota in the injurious behaviour of hens. The area is very current and as such there is a need for evaluation of the literature. At current, however, the work is quite confusingly written, and I am not confident that the literature has been summarised objectively. I have provided some specific comments on the PDF version of the manuscript but please see the following key areas which must be addressed for future versions of the manuscript.

  1. Paper type. You have labeled this paper as a review, yet it contains what appears to be a research project. To be clear, is this a previously published project? if so, please refrain from covering your own paper in detail as this confuses the reader; a review should summarise the key literature in a topic area. Upon reviewing your paper I was unclear as to why the paper's results were reported in your review, especially when much of the methods had not been explained.
  2. Bacillus. You have focused in on Bacillus, yet it is not clear why Bacillus has been selected. This is also not reflective of your title, which covers gut microbiota in general. Either adjust your title to reflect this, or the paper, ensuring you consider the full range of probiotics and their effects. Similarly, you may want to explain the physiological effects of Bacillus in more detail, especially as this is central to your review.
  3. Papers. How were papers sourced? You may want to provide an explanation as to how papers were found including search criteria used. 
  4. References and citations. There are many errors present throughout the citations and references, which are not fully formatted to the Animals MDPI style. Please check through all references carefully, as a range of errors are present.

There is some merit to this paper but there needs to be a much clearer message, and less focus on what is presumably the authors' previous papers. Cover what appears in the literature more objectively and consider where future researchers should investigate next - this is one of the key roles of a review.

Author Response

  1. Paper type. You have labeled this paper as a review, yet it contains what appears to be a research project. To be clear, is this a previously published project? if so, please refrain from covering your own paper in detail as this confuses the reader; a review should summarise the key literature in a topic area. Upon reviewing your paper, I was unclear as to why the paper's results were reported in your review, especially when much of the methods had not been explained.

Agree with the reviewer, the manuscript could be as a perspective article also. Based on the requirements, we performed systematic research on gut microbiota (microbiome) data in PubMed, SCOPUS, and EMBASE databases, and used Cochrane, ROBIN-I, and SYRCLE tools to assess the potential bias risks. The key words were used: gut microbiota (microbiome) composition, diversity, metabolite, and associations with functions in social stress-induced physiology, immunology, endocrinology, neurology, psychiatry, and behavior including fear, aggression, violence, feather pecking, and injury in animals including mammals, such as humans and rodents, and non-mammals, such as chickens. The manuscript was written based on the outcomes with the aim: if probiotics can reduce social stress and stress-related injurious behaviors in chickens,

  1. Bacillus. You have focused in on Bacillus, yet it is not clear why Bacillus has been selected. This is also not reflective of your title, which covers gut microbiota in general. Either adjust your title to reflect this, or the paper, ensuring you consider the full range of probiotics and their effects. Similarly, you may want to explain the physiological effects of Bacillus in more detail, especially as this is central to your review.

Agree with the reviewer, the title has been changed as “The impact of Bacillus subtilis-based probiotic on injurious behavior in laying hens.

  1. Papers. How were papers sourced? You may want to provide an explanation as to how papers were found including search criteria used.

Please see the response 1.

  1. References and citations. There are many errors present throughout the citations and references, which are not fully formatted to the Animals MDPI style. Please check through all references carefully, as a range of errors are present.

Sorry for the mistake. It has been changed.

There is some merit to this paper but there needs to be a much clearer message, and less focus on what is presumably the authors' previous papers. Cover what appears in the literature more objectively and consider where future researchers should investigate next - this is one of the key roles of a review.

Agree with the reviewer, and we were trying to include much broad articles, especially the studies related to the gut microbiota function in regulation of social stress-and stress-related behavior in poultry. Unfortunately, few studies were conducted in this area in chickens, and the most of the published articles are from our studies. As the reasons mentioned above, the manuscript was prepared with inclusion the findings from the patients with mental disorders. The aim of this special articles is to gain readers’ attention to the novel strategy, use of probiotics to modify gut microbiota composition, to reduce or prevent injurious damage in poultry; and to provide a new sight into the development of alternatives to beak trimming to improve health and welfare in laying hens.

Round 2

Reviewer 2 Report

Dear Authors,

Thank you for submitting a revised version of your manuscript. Whilst you have considered some of the feedback provided, it is clear that the larger concerns regarding the work have not been considered. For example, there is no explanation of your calm gentle birds or how they fit.

Ultimately, this is still not a review and there is still the reporting of research results which seem to have been previously published. This is of course a concern as the paper does not achieve what it set out to (i.e. review the field) and is also therefore biased toward the researcher's own publications. This paper therefore needs either to be formatted entirely into a review, or reformatted into a research paper that reports novel (not already published) findings.

Author Response

1.This is still not a review and there is still the reporting of research results which seem to have been previously published. This is of course a concern as the paper does not achieve what it set out to (i.e. review the field) and is also therefore biased toward the researcher's own publications. This paper therefore needs either to be formatted entirely into a review or reformatted into a research paper that reports novel (not already published) findings.

We are grateful for the reviewers’ comments and suggestions, and the chapters 4 and 5 have been rewritten.